# Predictors of development of cardiac and digestive disorders among patients with indeterminate chronic Chagas Disease

Erika Alessandra Pellison Nunes da Costa[1], Cassiano Victória[2], Carlos Magno Castelo Branco Fortaleza[1]*

1 Department of Infectious Diseases, Botucatu Medical School, São Paulo State University (UNESP), City of Botucatu, Brazil, 2 Department of Zoonosis, Faculty of Veterinary Medicine and Animal Science, São Paulo State University (UNESP), City of Botucatu, Brazil

* carlos.fortaleza@unesp.br

## Abstract

American trypanosomiasis (Chagas disease, CD) affects circa 7 million persons worldwide. While of those persons present the asymptomatic, indeterminate chronic form (ICF), many will eventually progress to cardiac or digestive disorders. We studied a nonconcurrent (retrospective) cohort of patients attending an outpatient CD clinic in Southeastern Brazil, who were admitted while presenting the ICF in the period from 1998 through 2018 and followed until 2019. The outcomes of interest were the progression to cardiac or digestive CD forms. We were also interested in analyzing the impact of Benznidazole therapy on the progression of the disease. Extensive review of medical charts and laboratory files was conducted, collecting data up to year 2019. Demographics (upon inclusion), body mass index, comorbidities (including the Charlson index) and use of Benznidazole were recorded. The outcomes were defined by abnormalities in those test that could not be attributed to other causes. Statistical analysis included univariate and multivariable Cox regression models. Among 379 subjects included in the study, 87 (22.9%) and 100 (26.4%) progressed to cardiac and digestive forms, respectively. In the final multivariable model, cardiac disorders were positively associated with previous coronary syndrome (Hazzard Ratio [HR], 2.42; 95% Confidence Interval [CI], 1.53–3.81) and negatively associated with Benznidazole therapy (HR, 0.26; 95%CI, 0.11–0.60). On the other hand, female gender was the only independent predictor of progression to digestive forms (HR, 1.56; 95%CI, 1.03–2.38). Our results point to the impact of comorbidities on progression do cardiac CD, with possible benefit of the use of Benznidazole.

## Author summary

Chagas Disease (CD) is a chronic, neglected infectious disease that affects several low-to-middle income countries. Besides its usual vector transmission, the etiological agent (*Trypanosoma cruzi*) can be transmitted through blood transfusion, so that migration from individuals with CD to Europe and North America. The scarcity and lack of evidence for

---

**Funding:** The authors received no specific funding for this work.

**Competing interests:** The authors have declared that no competing interests exist.

therapeutic options is a major challenge for clinical approach of CD patients. Most of those patients are diagnosed while in asymptomatic, indeterminate chronic form (ICF). Our study aimed at identifying factors associated with progression from ICF to cardiac or digestive disorders. We studied a cohort of 379 IFC patients in inner Brazil, of which 87 and 100 developed cardiac and digestive disorders. In univariate and multivariable analyses, the use of Benznidazole (one of the few drugs used for CD therapy) was statistically protective for progression to cardiac, but not to digestive CD forms. These findings emphasize the importance of novel research aimed at developing more effective therapeutic options.

## Introduction

Presently there are *circa* 7 million persons infected by *Trypanosoma cruzi*, the agent of Chagas disease (CD) [1]. In Brazil, especially in the Northern macro-region vector-borne disease transmitted by *Triatomineae* ("kissing bugs"), CD has been associated with poor housing conditions in rural areas [2]. However, that form of transmission has decreased as quality of housing improved in the country. However, the oral transmission (associated with ingesting the vector smashed alongside with açaí or sugarcane juices) has been increasingly reported. CD may also be transmitted through blood transfusion and vertically. Thus, European and North American countries are also at risk due to migration from endemic areas [1,3–6].

Most infected persons present the asymptomatic, indeterminate chronic form (ICF). However, up to 40% may progress over the years to cardiac or digestive forms [7,8]. The predictors for progression are still poorly understood, and the beneficial impact of specific anti-parasitic therapy with Benznidazole is not supported by robust evidence [9–14].

With that in mind, we studied a cohort of patients with CD (ICF) admitted over two decades to an outpatient clinic in inner São Paulo State, Brazil. Our objective was to identify predictors of development of localized forms of CD. We also attempted to analyze the impact of Benznidazole therapy in the prevention of progression to cardiac and/or digestive forms.

## Methods

### Ethics statement

This study was approved by the Committee for Ethics in Human Research from Faculdade de Medicina de Botucatu (project number CAAE: 65552917.8.0000.5411 / 1.999.171) in April 04th, 2017. Due to the retrospective collection of data, and according to Brazilian rules for human research, the study was exempt for obtention of formal consent.

### Study setting

The study was conducted in the Tropical Diseases outpatient service in Botucatu Medical School, São Paulo State University (UNESP). This clinic is linked to the university teaching hospital, and cares for patients in an area comprising approximately 1 million inhabitants from several municipalities surrounding Botucatu City (22° 53′ 25″ S, 48° 27′ 19″ W), inner São Paulo State, Brazil.

### Study design and subjects

We conducted a non-concurrent cohort, enrolling patients with CD who were admitted while presenting the ICF in the period from year 1998 through 2018. Even though inclusion period

ended in 2018, the follow-up period was extended up to December 2019. Even though we used a convenient sample of all outpatients meeting inclusion criteria, we performed a *post hoc* analysis of study power using OpenEpi software (Emory University, Atlanta, GA). We estimated the study power based on the impact of Benznidazole use and found it to be 89.7% and 96.1% for association with progression to cardiac and gastrointestinal CD.

## Operational aspects

The CD diagnosis was based on positivity in two serological tests (ELISA and indirect immunofluorescence). Only patients who were positive in both were included in this study. **Table 1** presents the protocol for clinical assessment of patients in the first medical consultation and during follow up. Those who presented abnormalities that could be attributed to either cardiac or digestive forms of CD upon admission were excluded from our study. However, we included subjects who presented heart diseases that could be attributed to coronary syndrome with or without myocardial infarction. ~çThe impact of that methodological decision on results and study limitations will be addressed in the discussion section.

We also excluded patients lost to follow up before year 2019. For the purpose of the present study, our inclusion criterium was the admission with ICF, defined as "serological positivity for CD, without presenting any cardiac or digestive abnormalities that could be attributed to that disease".

The clinical assessment presented in **Table 1** was used for exclusion of localized CD (e.g., for cardiac CD, dilated cardiomyopathy, congestive heart failure, arrhythmias, cardiac embolism, thromboembolic events; for digestive disease, megaesophagus, megacolon or motility disorders). However, we did include patients with heart diseases such as coronary syndrome not associated with dilated cardiomyopathy, congestive heart failure or arrhythmia). Of note, during follow-up, the attendant doctors registered diseases that affected study subjects, such as: myocardial infarction, systemic arterial hypertension, diabetes mellitus, cerebrovascular diseases, and other comorbidities, thus providing a list of "exposure factors" that were used in our cohort study. Benznidazole was used when indicated by the attendant doctor, based on the Brazilian Ministry of Health recommendations [15], which were the exclusion of localized forms (a condition for entrance to our cohort), age up to 50 years and manifested interest in receiving the therapy. When indicated, it was started as soon as the diagnosis of ICF of CD was characterized. The posology was 5-7mg/kg/day, divided in two doses, for 60 days. The outcomes of interest for our study (defined as abnormalities in the periodic tests, regardless of presenting symptoms) were also recorded in medical charts.

**Table 1.  Clinical and imaging assessment of patients presenting the indeterminate form of Chagas Diseases upon first medical consultation and during follow up.**

| Assessment | 1st consultation | Yearly | Upon medical request* |
|---|---|---|---|
| Thorough anamnesis | Yes | Yes | Not applicable |
| Physical examination | Yes | Yes | Yes |
| Electrocardiogram (ECG) | Yes | Yes | Yes |
| Chest radiography | Yes | Yes | Yes |
| Radiography of Esophagus Stomach Duodenum (with contrast) | Yes | No | Yes |
| Colon radiography (opaque enema) | Yes | No | Yes |
| Echocardiogram | No | Bo | Yes |

*Imaging of gastrointestinal tract was performed each time the patient reported even light symptoms odynophagia or constipation. Ecocardiogram was performed for all patients who presented dyspnea, limb edemas, or any sign or symptom suggestive of cardiac failure. Ecocardiogram was also performed in all patients in whom a novel ECG abnormality was identified. It is worth noting that patients with suspected cardiac or digestive symptoms/signs were frequently re-evaluated at intervals shorter than one year.

### Data collection and analysis

Extensive review of medical charts and laboratory files were conducted. Demographic data, comorbidities (defined by the International Classification of Diseases, 10[th] revision [ICD-10]) [15] and the Charlson comorbidity index [16] were recorded We also investigated the use of Benznidazole therapy. Data were analyzed in univariate and multivariable models of Cox regression, investigating time until development of the first localized form diagnosed (either cardiac or digestive). Variables were included gradually in multivariable models using a step-wise forward strategy, i.e., according to a crescent order of p-values [17]. A final value of *P*<0.05 was used as criterium for entry permanence in the models. No variable was forced in the models. All data analyses were performed using SPSS 20 (IBM, Armonk, NY).

## Results

A total number of 430 patients were diagnosed as presenting ICF in the first medical consultation, but 51 were lost to follow-up in 2019. Our study enrolled the remaining 379 subjects, of whom 87 (22.9%) and 100 (26.4%) progressed to cardiac and digestive forms, respectively. It is worth reporting that 10 patients presented both forms and were analyzed for the first of either outcome. A total of 192 (53.9%) subjects remained in the ICF through the total follow up period. Benznidazole was prescribed for 99 subjects, but 30 among them interrupted treatment in the first ten days due to adverse events, especially severe pruritus. All the other 69 patients completed 60-day therapy and were classified as "treated with Benznidazole" in our study.

The baseline characteristics of the cohort and of those who progressed to the outcomes of interest is presented in **Table 2**. As additional information, the findings that defined the progression to cardiac and digestive forms are presented in **Tables 3** and **4.**

Results from univariate and multivariable models of predictive for outcomes are presented in **Tables 5** and **6.** Briefly, in the final multivariable models, cardiac disorders were positively associated with previous myocardial infarction (Hazzard Ratio [HR], 2.42; 95% Confidence Interval [CI], 1.53–3.81) and negatively associated with Benznidazole therapy (HR, 0.26; 95% CI, 0.11–0.60). Female gender was the only independent predictor of progression to digestive forms (HR, 1.56; 95%CI, 1.03–2.38).The Kaplan-Meier curves for association of Benznidazole therapy with progression from ICF to cardiac or digestive forms are presented in **Figs 1** and **2.**

## Discussion

Even though urbanization, improvement of housing conditions and effective public preventive policies were introduced in the past two decades in Brazil [18], CD still threats a substantial proportion of the population, either because of new, orally-acquired disease and of the burden of cardiac and digestive sequelae [19]. This justifies the continuous interest in the epidemiology, clinical aspects, and therapy. There is special concern with the 1 to 2 million CD-affected persons who present the ICF [20]. Those are asymptomatic and will be diagnosed only through active search in population surveys and among blood donors. It is of utmost importance to reinforce the identification of asymptomatic CD patients, and to investigate factors predisposing to development of cardiac of digestive forms, including potential impact of therapeutic intervention [21]. Previous studies estimated that up to 30% ICF patients will develop one of those localized during their lifetimes [22]. Among our study subjects, that proportion was even greater, reaching 46.7%.

Comorbidities resulting from aging in individuals with CD as well as in the general population, have been verified not only by previous mortality studies, but also in cohorts of cases followed for long periods of time [23–25]. In our study, systemic arterial hypertension was present in 30% of the individuals and in 73.5% of the individuals with the cardiac form of the

**Table 2. Characteristics of Individuals with Cardiac and Digestive form of Chagas Disease.**

| Patients' characteristics | Total baseline cohort (379) | Patients who developed cardiac form (87) | Patients who developed digestive form (100) |
|---|---|---|---|
| Male gender | 188 (49.6) | 51(58.6) | 43(44.3) |
| Age, median (quartiles) | 49 (43–57) | 51(44–58) | 50(43–57) |
| Living in rural áreas | 116 (30.5) | 30(34.5) | 32(33.0) |
| Working as farmer | 131 (34.6) | 33(37.9) | 35(36.1) |
| Years of Schooling | | | |
| *0-4years* | 132 (34.8) | 33(37.9) | 40(41.2) |
| *5-8years* | 220 (58.0) | 50(57.5) | 51(52.6) |
| *>8years* | 27 (7.1) | 4(4.6) | 6(6.2) |
| Follow-up time [months], median (quartiles) | 120(72–168) | 84(48–132) | 96(60–156) |
| Therapy with Benznidazole | 69 (18.2) | **6(8.69)** | 24(24.7) |
| Body Mass Index [kg/m2], median (quartiles) | 27 (25–30) | 27.0(25.0–31.0) | 27(24.6–30.0) |
| Hypertension | 216(57) | **64(73.46)** | 5.5(53.6) |
| Diabetes | 79(20.80) | 24(27.6) | 17(17.5) |
| Dyslipidemia | 206(54.20) | 51(58.6) | 54(55.7) |
| Heart Disease | 56(14.70) | **28(32.2)** | 12(12.4) |
| Lung Disease | 46(12.10) | 15(17.2) | 11(11.3) |
| Kidney Disease | 10(2.60) | 4(4.6) | 0(0.0) |
| Liver Disease | 9(2.40) | 3(3.4) | 1(1.0) |
| Neurovascular Disease (Stroke) | 24(6.30) | **11(12.6)** | 5(5.2) |
| SolidMalignancy | 39(10.30) | 12(13.8) | 6(6.2) |
| Lymphoma/Leukemia | 2(0.5) | 1(1.1) | 1(1.0) |
| AIDS | 1(0.30) | 1(1.1) | 0(0.0) |
| Diverticular disease | 85(22.40) | 13(14.9) | 25(25.8) |
| Charlson Comorbidity Index, median (quartiles) | 1 (1–2) | **2(1–3)** | 1(1–2) |
| Steroids | 2(0.5) | 0(0.0) | 1(1) |
| Tyroid diseases | 71(18.70) | 22(25.3) | 17(17.5) |
| Vascular diseases | 39(10.30) | 14(16.1) | 8(8.2) |

**Note.** Variables presented in N (%), except: age, follow-up, Body Mass Index and Charlson's Comorbidy Index, which are presented in median (quartiles).Statically significant (p<0.05) differences from the baseline cohort are presented in boldface.

**Table 3. Cardiologic findings of Chagas Disease in routine exams.**

| Eletrocardiogram | | Echocardiography | | |
|---|---|---|---|---|
| | | No VD | LVDD | LVSD |
| RBBB | 16 | 13 | 2 | 1 |
| LBBB | 24 | 19 | 4 | 1 |
| RBBB+LBBB | 14 | 10 | 3 | 1 |
| AVB | 16 | 9 | 4 | 2 |
| AF | 4 | 0 | 4 | 1 |
| DC | 13 | 0 | 8 | 5 |
| Total | 87 | 51 | 25 | 11 |

RBBB, Right bundle branch block; LBBB, left bundle branch block; AVB, atrioventribular block; AF, atrial fibrillation; DC, dilated cardiomyopathy.

**Table 4. Digestive exams in patients of Chagas Disease.**

| Imaging | Digestive Organ | Patients with abnormalities |
|---|---|---|
| ESD | Esophagus | |
| | Megaesophagus, grade 1 | 26 |
| | Megaesophagus, grade 2 | 13 |
| | Megaesophagus, grade 3 | 6 |
| | *Subtotal* | 45 |
| OE | Colon | |
| | Sigmoid | 33 |
| | Transverse | 10 |
| | Ascending | 4 |
| | Rectum | 8 |
| | *Subtotal* | 55 |
| | Total | 100 |

ESD,Radiography of Esophagus Stomach Duodenum, with contrast; OE, Opaque enema.

**Table 5. Factors associated with progression from chronic indeterminate to cardiac form of Chagas Disease.**

| | Univariate analysis | | Multivariable Analyis | |
|---|---|---|---|---|
| Risk factors | HR (95%CI) | P | HR (95%CI) | P |
| Male gender | 1.24 (0.80–1.91) | 0.33 | **1.03(1.01–1.05)** | **0.009** |
| Age, median (quartiles) | **1.03(1.01–1.05)** | **0.004** | | |
| Living in rural área | 1.24 (0.80–1.94) | 0.34 | | |
| Working as a farmer | 1.19 (0.77–1.84) | 0.43 | | |
| Years of schooling | | | | |
| *0–4 years* | | | | |
| *5–8 years* | 0.81 (0.52–1.27) | 0.36 | | |
| *>8years* | 0.55(0.20–1.56) | 0.26 | | |
| Use of Benznidazole | **0.21 (0.09–1.07)** | **<0.001** | **0.26(0.11–0.60)** | **0.002** |
| Body Mass Index (kg/m$^2$) | 1.02(0.98–1.07) | 0.35 | | |
| Hypertension | 2.13(1.32–3.44) | **0.002** | | |
| Diabetes | 1.35(0.84–2.16) | 0.22 | | |
| Dyslipidemia | 1.07(0.70–1.64) | 0.76 | | |
| Heart Disease* | **2.77(1.76–4.36)** | **<0.001** | **2.42(1.53–3.81)** | **< 0.001** |
| Lung Disease | **1.97 (1.12–3.46)** | **0.02** | | |
| Kidney Disease | 2.19(0.80–5.99) | 0.13 | | |
| Liver Disease | 1.26(0.40–3.99) | 0.69 | | |
| Neurologic Disease | **2.79(1.48–5.26)** | **0.002** | | |
| Solid Tumor | 1.56(0.85–2.89) | 0.15 | | |
| Lymphoma/Leukemia | 3.28(0.46–2.70) | 0.24 | | |
| AIDS | 4.69(0.65–3.84) | 0.13 | | |
| Diverticular disease | 0.59(0.33–1.05) | 0.07 | | |
| Charlson comorbidity Index, median (quartiles) | **1.38 (1.19–1.59)** | **<0.001** | | |
| Use of Steroids | 0.04(0.0-. . . .) | 0.66 | | |
| Thyroid diseases | **1.75 (1.08–2.85)** | **0.02** | | |
| Vascular Disease | 1.39(0.78–2.48) | 0.27 | | |

Note. Statistically significant results (*P*<0.05) are presented in boldface. *Heart diseases not attributed to Chagas Disease were restricted to coronary syndrome, with or without myocardial infarction.

**Table 6. Factors associated with progression from chronic indeterminate to digestive form of Chagas Disease.**

| Risk Factors | Univariate Analysis | | Multivariable Analysis | |
|---|---|---|---|---|
| | HR(95%CI) | *P* | HR (95%CI) | *P* |
| Male gender | **0.65 (0.43–0.99)** | **0.04** | **0.65 (0.43–0.99)** | **0.04** |
| Age, median (quartiles) | 1.01(0.99–1.03) | 0.2 | | |
| Living in rural área | 1.10(0.71–1.70) | 0.68 | | |
| Working as a farmer | 1.05(0.69–1.61) | 0.81 | | |
| Years of schooling | | | | |
| *0–4 years* | | | | |
| *5–8 years* | 0.66(0.43–1.00) | 0.051 | | |
| *>8 years* | 0.67 (0.28–1.58) | 0,36 | | |
| Use of Benznidazole | 0.88(0.55–1.42) | 0.59 | | |
| Body Mass Index (kg/m$^2$) | 1.00(0.96–1.05) | 0.88 | | |
| Hypertension | 0.92(0.62–1.39) | 0.70 | | |
| Diabetes | 0.78(0.46–1.32) | 0.35 | | |
| Dyslipidemia | 0.95 (0.63–1.43) | 0.8 | | |
| Heart Disease* | 0.87(0.47–1.60) | 0.65 | | |
| Lung Disease | 1.37(0.71–2.52) | 0.37 | | |
| Kidney Disease | 0.05 (0.00–28.13) | 0.35 | | |
| Liver Disease | 0.38(0.05–2.71) | 0.33 | | |
| Neurologic Disease | 0.89(1.33–2.44) | 0.83 | | |
| Solid Tumor | 0.69 (0.30–1.57) | 0.37 | | |
| Lymphoma/Leukemia | 3.47(0.48–25.06) | 0.22 | | |
| AIDS | 0.05(0.00-. . . .) | 0.75 | | |
| Diverticular disease | 1.16(0.94–1.44) | 0.17 | | |
| Charlson comorbidity Index, median (quartiles) | 0.85 (0.69–1.05) | 0.14 | | |
| Use of Steroids | 2.68(0.37–19.30) | 0.33 | | |
| Thyroid diseases | 1.06(0.62–1.082) | 0.83 | | |
| Vascular Disease | 0.66(0.32–1.37) | 0.27 | | |

Note: The factors were described through univariate and multivariate analysis,with a 95% confidence interval and p value (<0,05). *Heart diseases not attributed to Chagas Disease were restricted to coronary syndrome, with or without myocardial infarction.

CD. Data from literature agree with our findings regarding age and comorbidities [26]. Furthermore, studies measuring levels of kinins and nitric oxide suggested a synergistic activity of CD and systemic arterial hypertension in the myocardial remodeling process [27]. However, other studies found that *T. cruzi* infection did not alter the outcome of subjects with systemic arterial hypertension [28,29]. Furthermore, systemic arterial hypertension associated with CD myocardiopathy may be an important risk factor in the genesis of ischemic and hemorrhagic cerebrovascular diseases, as well as in life-threatening embolisms [30,31]. Those findings open interesting venues for future research. The other risk factor identified in our multivariable analysis (coronary syndrome), though not usually an exclusive consequence of CD, may have been a concurrent cause for cardiac abnormalities found during follow up [32].

In our study, 69 patients completed Benznidazole therapy, of whom 6 (8.7%) developed the cardiac form, while 81 out of 310 non-treated patients presented heart disorders. The protective effect of Benznidazole is evidenced by the Rate Ratio (RR) of 0.33 (95% Confidence Interval, 0.16–0.73, *P*<0.001). These findings mirror the Cox regression results presented in **Table 5** and are coherent with previous findings from observational studies [33,34].

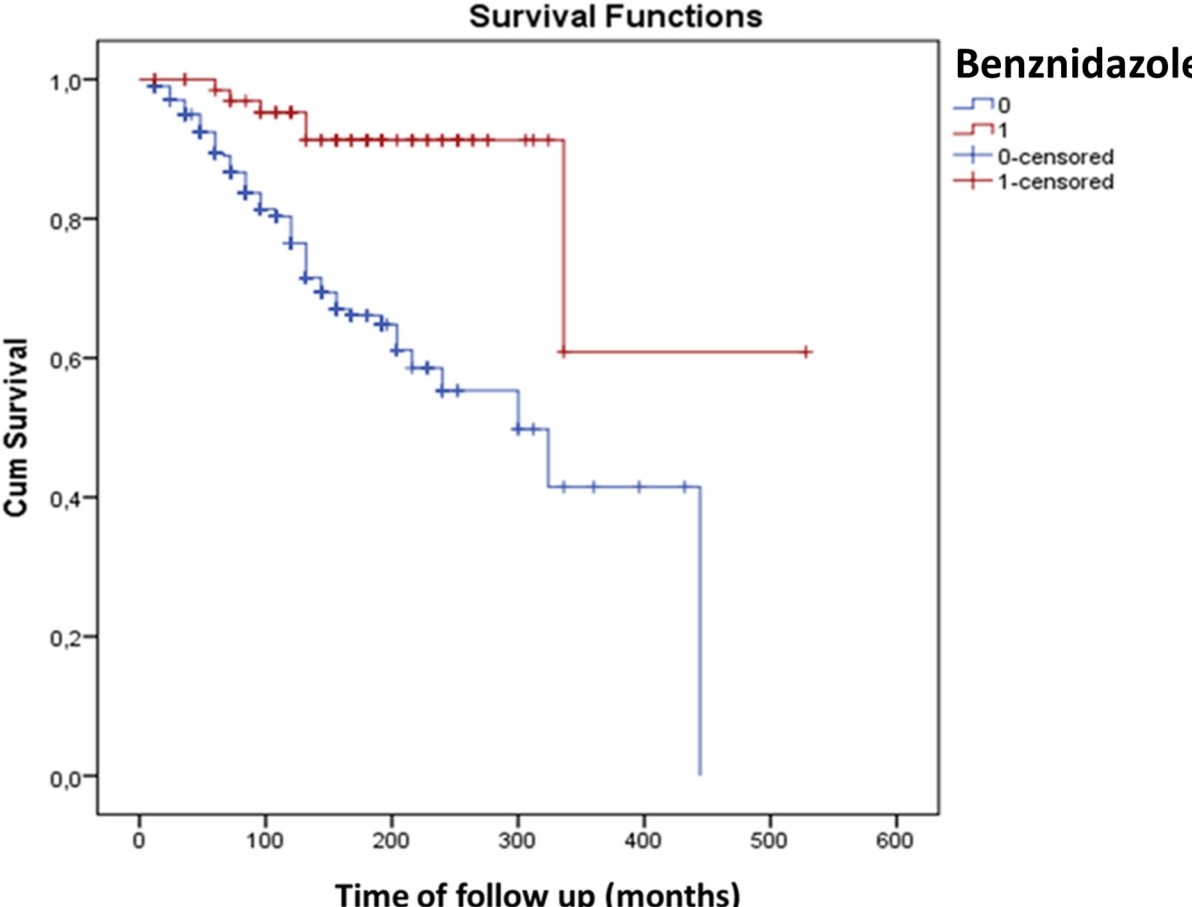

**Fig 1. Kaplan-Meier graphic for association of Benznidazole therapy and progression from the indeterminate to the cardiac form of Chagas Disease.** *Note*. Benznidazole use, 1 = yes; 0 = no. The Log-rank test for difference of curves was statistically significant (*P*<0.001).

Methodological concerns may be raised for the inclusion of "heart diseases" among predictors of progression to cardiac CD. As a counterfactual test, we repeated the analysis excluding from the baseline population both patients with "heart diseases" or with systemic hypertension. The alternative analysis included 163 subjects, of whom 36 22.1%) developed cardiac CD. Of note, none of those patients receiving Benznidazole progressed to cardiac CD, while 24 out of 127 subjects progressed to cardiac DC (18.9%). Results of the final multivariable Cox model are presented in **Table 7**, and Kaplan-Meier result for association with Benznidazole use is presented in **Fig 3**. When the same models tested not including the use of Benznidazole among independent variables, only lung disease (HR, 4.16; 95%CI, 1.51–11.49; *P* = 0.006) is associated with development of cardiac CD. Additionally, we conducted analysis including separately those patients with hypertension or with a previous cardiac disease (**Table 8**). The use of Benznidazole was negatively associated with progression to cardiac CD among subjects with systemic hypertension, but not among those with previous cardiac disorders (of whom only 5 were treated with Benznidazole).

The analysis of the sub-cohort patients without systemic hypertension or heart diseases also respond to a second methodological concert, which is misclassification bias in outcomes. It is true that disorders defining cardiac CD could arise from concurrent comorbidities or

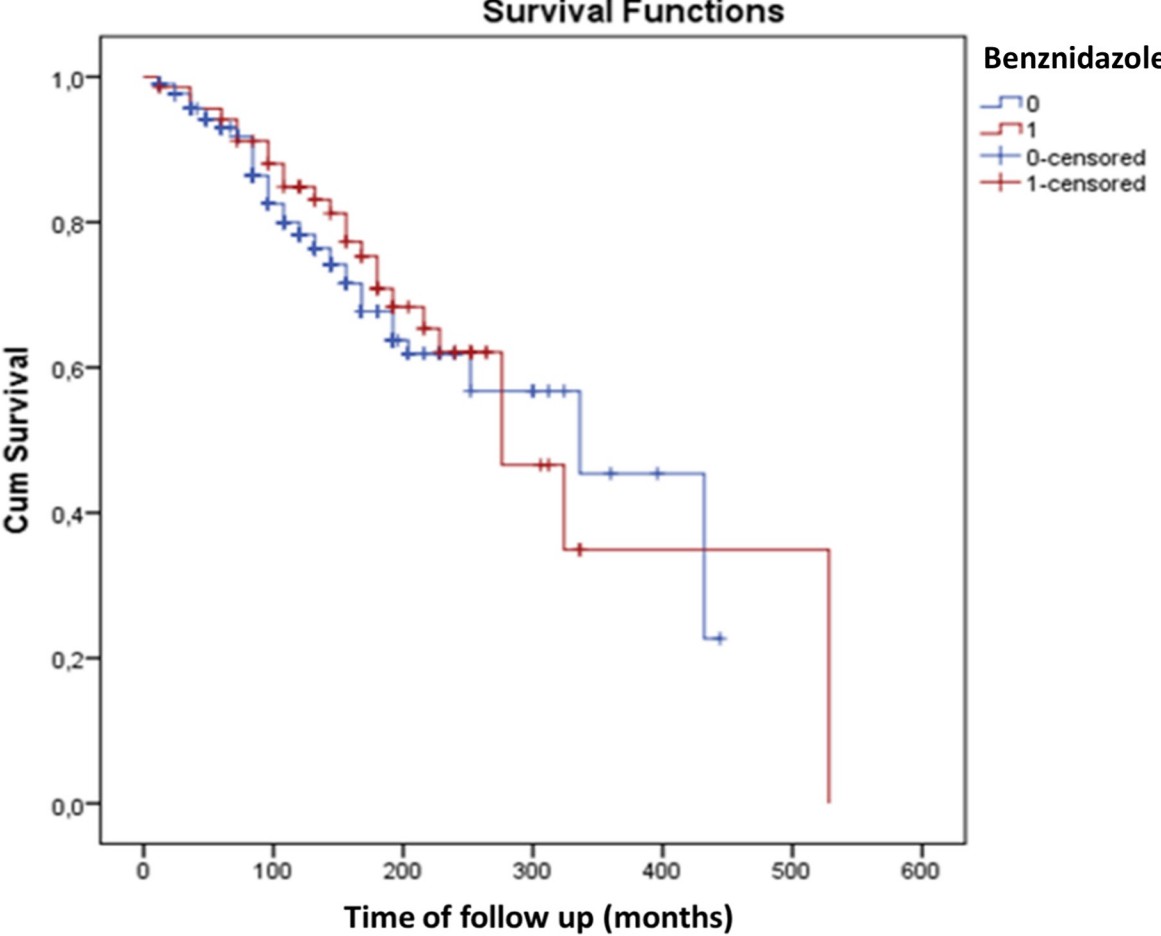

**Fig 2. Kaplan-Meier graphic for association of Benznidazole therapy and progression from the indeterminate to the digestive form of Chagas Disease.** *Note.* Benznidazole use, 1 = yes; 0 = no. The Log-rank test for difference of curves was statistically significant ($P = 0.59$).

environmental exposures. Still, the alternative analysis rules out the two major potential sources of misclassification bias.

Both our original and alternative models point to the role of comorbidities in the progression from ICF to cardiac form o CD. Those findings open relevant venues for further studies focusing on the pathophysiology of CD.

As a separate item, a possible protective association of the use of Benznidazole regarding progression to cardiac CD was detected. The most robust randomized clinical trial

**Table 7. Final multivariable Cox-regression results for predictors of progress from indeterminate chronic form to cardiac form of Chagas Disease, <u>excluding</u> patients who either presented other heart diseases or systemic arterial hypertension.**

| Risk factors | HC (95%CI) | P |
|---|---|---|
| Lung disease | 3.69 (1.41–10.64) | 0.009 |
| Use of Benznidazole | 0.09 (0.01–0.69) | 0.02 |

Note. The baseline cohort for this analysis included included 163 subjects, of whom 36 (22.1%) developed cardiac Chagas Disease

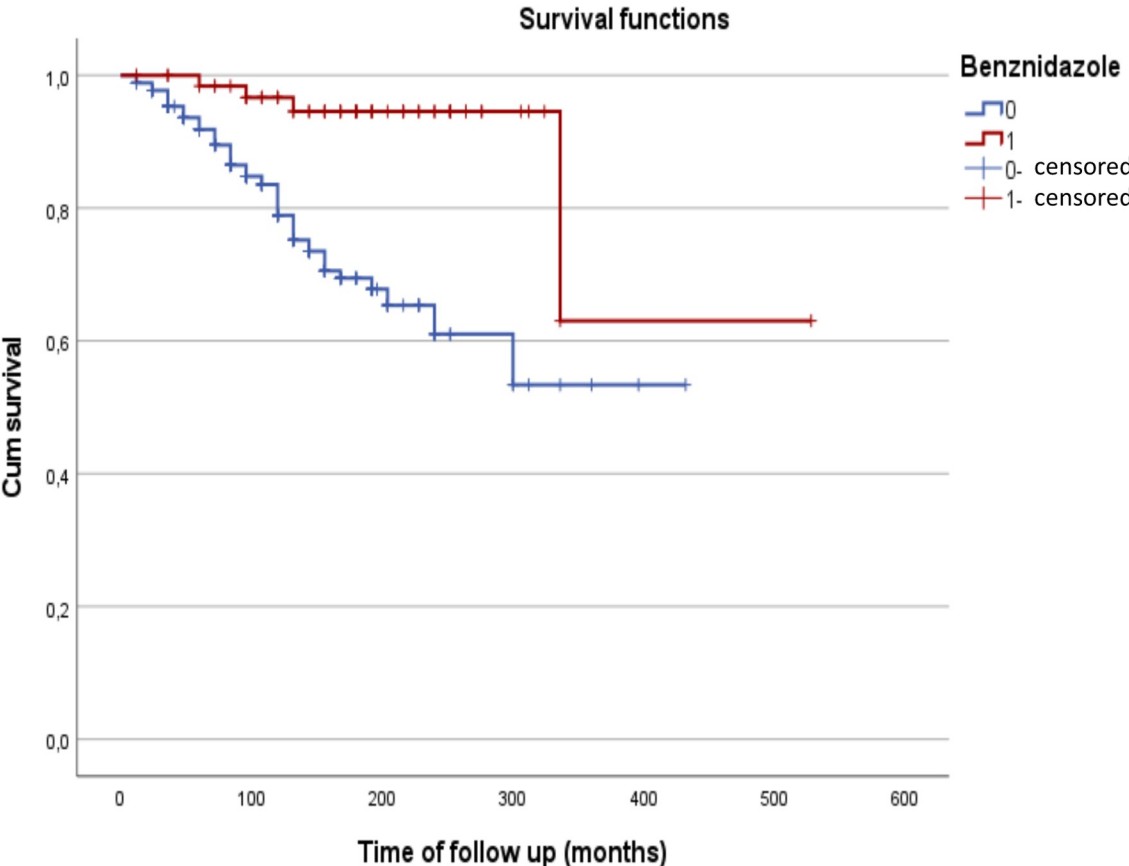

**Fig 3. Kaplan-Meier graphic for association of Benznidazole therapy and progression from the indeterminate to the cardiac form of Chagas Disease, with a baseline population excluding patients with other heart diseases or systemic hypertension.** Note. The baseline cohort for this analysis included 163 subjects, of whom 36 (22.1%) developed cardiac Chagas Disease. Benznidazole use, 1 = yes; 0 = no. The Log-rank test for difference of curves was statistically significant (P<0.001).

(BENEFIT), which randomized 2854 patients with mild cardiac form for use of Benznidazole versus placebo, found that patients who were treated with Benznidazole presented clearance of the parasite (as detected by polymerase chain reaction [PCR]). However, the therapy did not

**Table 8. Final multivariable Cox-regression results for predictors of progress from indeterminate chronic form to cardiac form of Chagas Disease, <u>including</u> patients who either presented other heart diseases or systemic arterial hypertension.**

| Risk factors | HC (95%CI) | P |
| --- | --- | --- |
| **Only patients with hypertension** | | |
| Central Nervous System Disease | 2.31 (1.14–4.71) | 0.02 |
| Use of Benznidazole | 0.40 (0.17–0.94) | 0.03 |
| **Only patients with other heart diseases** | | |
| Renal disease | 5.20 (1.40–23.67) | 0.03 |
| Use of Benznidazole | 0.32 (0.54–2.35) | 0.42 |

**Note.** Baseline cohort for patients with hypertension included 216 subjects, of whom 33 (15.2%) were treated with benznidazole and 64 (29.6%) progressed to cardiac Chagas Diseases. The cohort including patient with other heart diseases included 56 patients, of whom 5 (8.9%) were treated with benznidazole and 28 (50.0%) progressed to our criteria for defining cardiac Chagas Disease.

prevent progression to cardiac over 5 years [35]. Even though patients included in the study already presented mild abnormalities, this was the greatest clinical trial addressing the impact of anti-parasitic therapy on the progression of cardiac forms. Interestingly, a preventive impact on that progression was found in subgroup analysis including only patients from Brazil [36].

Our analysis of predictors of progress from ICF to digestive CD also extensively assessed demographics and comorbidities. However, only female gender was a predictor of progression in Cox models. Also, as expected from previous studies [37], digestive complications of CD were not prevented by the use of Benznidazole in our cohort study.

Besides neglection by pharmaceutical industry, CD therapy studies are hampered by difficulties in outcome definitions (e.g., parasitological, molecular, clinical). Presently we cannot assure that lowering the blood parasite counts or achieving negative PCR implies non-progression to either cardiac or digestive forms. Furthermore, clinical trials require term follow up, which can lead to negative results, as exemplified by the BENEFIT study [36]. Since Benznidazole can cause serious adverse effects, its recommendation requires extreme care, and strengthening of current evidence [38]. This reinforces the importance of cohort studies with data collected over an extensive period.

Our study is limited by the non-concurrent design. Also, there is a relevant possibility of misclassification bias (both in exposures and outcomes) occurred, and this limits conclusions such as the benefit of Benznidazole for preventing cardiac CD. We attempted to overcome those flaws with both multivariate models and several alternative analysis as counterfactuals to our results. We do believe there is an important strength in our study: patients were followed with a rigorous clinical protocol to identify both exposures and outcomes. Finally, we performed extensive chart review and robust statistical analysis.

In conclusion, the progression for ICF to cardiac CD is associated with comorbidities (such as lung, central nervous system, and coronary syndrome). Though biases might have occurred (se above), we found Benznidazole to be protective for development of CD in most analyses. That finding should be tested in new clinical trials. On the other hand, we did not find impact of comorbidities or therapy with Benznidazole on the progression to digestive CD.

In a separate topic, our findings reinforce the importance of a periodic assessment of ICF patients with electrocardiograpy and contrasted gastrointestinal imaging, as presented in Table 1. Performing active search for identification and possibly therapy of asymptomatic CD patients may be a wise strategy to lessen the burden of cardiac sequelae in low-to-middle income countries.

## Supporting information

**S1 Strobe checklist.**
(DOCX)

**S1 Data. Anonymized database containing data from which our analysis was performed, included as supplementary file in a Microsoft Excel Spreadsheet.**
(XLSX)

## Author Contributions

**Conceptualization:** Erika Alessandra Pellison Nunes da Costa, Cassiano Victória, Carlos Magno Castelo Branco Fortaleza.

**Data curation:** Erika Alessandra Pellison Nunes da Costa, Cassiano Victória, Carlos Magno Castelo Branco Fortaleza.

**Formal analysis:** Erika Alessandra Pellison Nunes da Costa, Cassiano Victória, Carlos Magno Castelo Branco Fortaleza.

**Investigation:** Erika Alessandra Pellison Nunes da Costa, Cassiano Victória, Carlos Magno Castelo Branco Fortaleza.

**Methodology:** Erika Alessandra Pellison Nunes da Costa, Cassiano Victória, Carlos Magno Castelo Branco Fortaleza.

**Project administration:** Erika Alessandra Pellison Nunes da Costa, Cassiano Victória, Carlos Magno Castelo Branco Fortaleza.

**Supervision:** Carlos Magno Castelo Branco Fortaleza.

**Validation:** Erika Alessandra Pellison Nunes da Costa, Cassiano Victória, Carlos Magno Castelo Branco Fortaleza.

**Visualization:** Erika Alessandra Pellison Nunes da Costa, Cassiano Victória, Carlos Magno Castelo Branco Fortaleza.

**Writing – original draft:** Erika Alessandra Pellison Nunes da Costa, Cassiano Victória, Carlos Magno Castelo Branco Fortaleza.

**Writing – review & editing:** Erika Alessandra Pellison Nunes da Costa, Cassiano Victória, Carlos Magno Castelo Branco Fortaleza.

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
