## [Decision Letter · Decision Letter 0]

7 Sep 2020

Dear Dr. Fortaleza,

Thank you very much for submitting your manuscript "Predictors of development of cardiac and digestive disorders among patients with chronic indeterminate Chagas Disease." for consideration at PLOS Neglected Tropical Diseases. As with all papers reviewed by the journal, your manuscript was reviewed by members of the editorial board and by several independent reviewers. In light of the reviews (below this email), we would like to invite the resubmission of a significantly-revised version that takes into account the reviewers' comments. 

We cannot make any decision about publication until we have seen the revised manuscript and your response to the reviewers' comments. Your revised manuscript is also likely to be sent to reviewers for further evaluation.

Sincerely,

Christine A Petersen

Deputy Editor

Christine Petersen

Deputy Editor

Reviewer's Responses to Questions

**Key Review Criteria Required for Acceptance?**

**Methods**

-Are the objectives of the study clearly articulated with a clear testable hypothesis stated?

-Is the study design appropriate to address the stated objectives?

-Is the population clearly described and appropriate for the hypothesis being tested?

-Is the sample size sufficient to ensure adequate power to address the hypothesis being tested?

-Were correct statistical analysis used to support conclusions?

-Are there concerns about ethical or regulatory requirements being met?

Reviewer #1: There is a need for important clarifications on study design and methodology. Though the design is described as non-concurrent design, it is not totally clear if the study was prospective or retrospective. It would be important to present and review the protocol of the study.

An important consideration for this type of study is the standardisation of assessments performed over time and the treatment of missing data. No detailed information is provided on any of the assessments for disease progression (echo, electrocardiographic readings, and evaluation of digestive manifestations) - it is unclear which parameters were to be tracked were defined a priori, if different assessors were involved, if there was independent review of information, what were the data collection methods, etc. Also there is no information on Benznidazole treatment, dose, compliance and assessment of response.

Reviewer #2: Objectives

- The study has the objective to determine predictors for the development of cardiac and digestive disorders among patients with chronic and indeterminate Chagas' Disease. The study's purpose is outstanding. Such determinants can enlighten clinicians on the particular variables that are most relevant for monitoring patients, and that can ascertain the conduct of the medical team. The findings would also have the potential to establish a new hypothesis to incentivize new research avenues for the study of Chagas' Disease.

- The central idea of the study's aim is to determine predictors for the development of cardiac and digestive disorders. However, on page 6, lines 1-2, the authors casually mention in a short sentence that they will also investigate the use of Benznidazole. Nevertheless, the authors have a great focus on the use of the drug in the "Results" section, and the majority of the "Discussion" section is used to address the use of this anti-parasitic drug. Therefore, there is a discrepancy between the study's goals and what the study ended up focusing on. The findings on this drug are relevant and should be described in the study. Still, the author should make it clear to the reader that the study has more than one aim: the first being determining predictors of cardiac and digestive disorders in Chagas' Disease patients; the second being if there is an association between Benznidazole and the outcomes of interest, and if this may suggest a protective effect. The same clarity should be provided in the "Introduction" section and the title of the manuscript.

Study design and population

- No mention is made in the manuscript to describe if the sample size sufficient ensures adequate power to address the hypothesis being tested. However, the study uses a convenient sample of all the patients with a Chagas' Disease diagnosis who were admitted while presenting the disease's indeterminate chronic form (ICF), and the study does yield statistical significance.

- The study is defined as a "non-concurrent cohort." Despite its description of outcomes, as the development of cardiac and digestive disorders, no clear distinction is made in the "Methods" section about what constitutes the "exposure" as it is inherent expected in a cohort study. The authors do not mention the "prospective" nature of the study, but the data collection is prospective and obtained from patients who enrolled between 1998-2018. 

- On Page 5, lines 9-10, the authors describe the research subjects as participants who were admitted with the indeterminate chronic form of Chagas' disease between the years 1998-2018, but that data was collected until 2019. To reduce the chances for confusion, I advise the authors to make it clearer for the readers that 1998-2018 was the window for enrollment of the recruited patients, but that data was obtained from beyond the enrollment window, and therefore, 2019 is also included.

- Inclusion criteria were defined as "patients with CD who were admitted while presenting the ICF in the period from year 1998 through 2018" (page 5, lines 8-10). Exclusion criteria were defined as "Those who presented abnormalities upon admission."

Statistical analyses

- Statistical analyses utilized univariate and multivariable Cox regression models, which could be considered appropriate to assess associations between particular variables and the outcomes of interest. As a suggestion, the author could have also used data mining to study predictive models of associations among the variables.

- The authors were unclear as to which variables were considered for each of the multivariable models. The authors mentioned that variables were included gradually in multivariable models using a stepwise forward strategy. However, no additional documents were provided with the statistical analyses to clarify the variables used for each measurement.

- The authors should describe in detail in the "Methods" section, which variables were included in each of their multivariable models. This practice refines the reader's understanding of which variables have an association with the outcomes of interest in others' presence. The authors should also provide additional documentation from their statistical analysis performed using SPSS 20.

- There are no mentions of possible bias, confounders or effect modifies that could have interfered with the analyses.

- The statistical analyses do not support all of the conclusions. In the "Author Summary" section, page 3, lines 11-12, the author claims that "Benznidazole (...) prevented the progression to cardiac, but not digestive CD forms." The current study cannot support the claim. The author can attest that the univariate Cox regression demonstrated a statistically significant association between the use of Benznidazole and the occurrence of cardiac disorder. Further studies would be necessary to attest to the efficacy of the drug as the manuscript currently described it.

Ethical perspectives

- The study does not presents concerns about ethical or regulatory requirements being met. It followed principles from the Helsinki declaration and was approved by their local Committee for Ethics in Human research. However, one of the attached documents presented by the authors reveals private information of participants, with their names and electronic record numbers. Recommendations to the authors are described in the "Summary and General Comments" section of this peer-review.

Reviewer #3: See attached review

**Results**

-Does the analysis presented match the analysis plan?

-Are the results clearly and completely presented?

-Are the figures (Tables, Images) of sufficient quality for clarity?

Reviewer #1: - No detailed information on analyses plan/protocol for the study. 

- Unclear how missing information on patients loss to follow-up were addressed for analyses. It would be useful to include sensitivity analyses.

Reviewer #2: - The authors presented an exciting data set that can be appreciated by researchers from different research fields of neglected tropical diseases, including clinical and epidemiological insights that can help theorize some plans for pathophysiology investigations.

- The analysis presented matches the analysis plan. Surprisingly, the results on Benznidazole's use receive a higher focus than initially suggested by the "Introduction" or "Methods." The results on this anti-parasitic are indeed relevant. However, the authors should make that clear in the earlier sections of the manuscript, including its title, that they intend to give this much focus on the drug.

- The "Results" section revealed that of the 87 subjects who develop cardiac complications, and 100 subjects who develop digestive complications, 10 participants developed both. Nevertheless, the subjects who developed both complications are left are participants of the two primary outcomes. Perhaps the authors should consider if these 10 individuals should constitute a separate outcome group, as their susceptibility may differ. That way, there could be four outcome groups: 1- No change from ICF; 2- Cardiac complications; 3- Digestive complications; 4- Both cardiac and digestive complications. If this is taken under consideration, then table 1 should be modified. In consideration of four outcome groups, the 10 individuals should also be removed from tables 4 and 5.

Considerations for table 4 and their associated results:

- Should the 10 individuals who also presented digestive changes be accounted for inside this group?

- "Age" is statistically significant but is not all in bold, as suggested by the table's footnote.

- "Hypertension" is statistically significant, but is not all in bold, as suggested by the table's footnote. There is no multivariable analysis demonstrated for "hypertension."

- "Lung Disease" is statistically significant, but is not all in bold, as suggested by the table's footnote. There is no multivariable analysis demonstrated for "Lung Disease." There is no mention of "Lung Disease" in the "Results" section's text on page 6.

- "Neurological Disease" is statistically significant, but is not all in bold, as suggested by the table's footnote. There is no multivariable analysis demonstrated for "Neurological Disease." There is no mention of "Neurological Disease" in the "Results" section's text on page 6.

- "Tyroid Disease" is statistically significant, but is not all in bold, as suggested by the table's footnote. There is no multivariable analysis demonstrated for "Tyroid Disease." There is no mention of "Tyroid Disease" in the "Results" section's text on page 6.

- "Charlson comorbidity index" is statistically significant, but is not all in bold, as suggested by the table's footnote. There is no multivariable analysis demonstrated for " Charlson comorbidity index." There is no mention of " Charlson comorbidity index " in the "Results" section's text on page 6.

- It is unclear if the authors only intended to perform multivariable analysis on the statistically significant variables from the univariate analysis. If that is the case, it is unclear why "hypertension","lung disease", "neurological disease", "tyroid disease" and "Charlson comorbidity index" had no multivariable analysis.

- It is unclear what are the variables that are considered in the multivariable analysis.

- It is unclear if confounders or effect modifiers were considered in the multivariable analyses, and if so, which ones.

Considerations for table 5 and their associated results:

- On page 6, lines 21-22, the authors say that "female gender was the only independent predictor of progression to digestive forms." However, table 5 reveals that "depression" was also an independent predictor. The authors fail to mention this relationship.

- It is unclear if the author only intended to perform multivariable analysis on the statistically significant variables from the univariate analysis. If that is the case, it is unclear why "depression" had no multivariable analysis.

- It is unclear what are the variables that are considered in the multivariable analysis.

- It is unclear if confounders or effect modifiers were considered in the multivariable analyses, and if so, which ones.

Considerations for figure 1 and their associated results:

- The survival functions described in figure 1 are related to the risk of developing cardiac complications (figure 1A) or digestive complications (figure 1B). The graph should be made clear, both in the title and the y-axis, to avoid confusion that the survival could be analyzing the risk of death.

- It would be interesting to know how the results from table 1 and the association with Benznidazole behaves when accounting for other variables, such as "age," "hypertension," "myocardial infarction," and other pre-existing cardiovascular conditions.

Reviewer #3: (No Response)

**Conclusions**

-Are the conclusions supported by the data presented?

-Are the limitations of analysis clearly described?

-Do the authors discuss how these data can be helpful to advance our understanding of the topic under study?

-Is public health relevance addressed?

Reviewer #1: Need to address/discuss in the conclusions quite a few issues and limitations:

• Important to clarify the design of the study (if retrospective versus prospective)y

• Standardisation of assessments or lack thereof 

• Ascertainment on treatment and low number of patients treated with Benznidazole

• Lack of serological and/or parasitological response markers

Reviewer #2: - The authors appropriately discuss the epidemiological importance of the study and how it particularly relates to Brazil, the country where the study has been conducted.

- The "Discussion" section has a disproportional emphasis on statistically significant association observed between the use of Benznidazole and the progression to cardiac forms of Chagas' Disease. Unlike what the authors initially claim will be the primary intent of the study, the discussion and conclusion focus on the use of the drug and only make a brief mention of one of the possible predictors of disease: "hypertension."

- The discussion and conclusion should be reworked to address the aims the manuscript initially sets.

- The last paragraph of the "discussion" section holds the claim that Benznidazole is beneficial for the prevention of progression of ICF to cardiac forms of Chagas Disease. However, this is not entirely supported by the data. The data only demonstrates a strong and statistically significant association between Benznidazole and the cardiac progression of Chagas' Disease. Although suggestive of a protective effect, not much was discussed on how other variables' presence may interfere with this association.

- The authors describe the limitation of the non-concurrent design of the study. Nevertheless, there could be a mention of other limitations, such as possible bias, confounders, and effect modifiers.

- The authors discuss how these data can help advance our understanding of the topic under study, but different than the initial aim of determining predictors of disease progression, it primarily focuses on Benznidazole's therapeutic potential.

- The authors present rich data and interesting results. Unfortunately, there are missed opportunities on result items that could or should have been addressed in the discussion. To name a few:

- The authors observe a statistically significant association between "age" and the development of cardiac problems. Despite the Chagas' disease diagnosis, age is usually associated with increased risk for cardiac complications. What hypothesis could be drawn? What could be discussed?

- The authors observe a statistically significant association between "hypertension" and the development of cardiac problems. Despite the Chagas' disease diagnosis, hypertension is usually associated with increased risk for cardiac complications, including cardiac failure, cardiac wall weakening, and cardiac hypertrophy. What hypothesis could be drawn? What could be discussed?

- The authors observe a statistically significant association between "myocardial infarction" and additional cardiac problems. Despite the Chagas' disease diagnosis, the occurrence of myocardial infarction is already associated with increased risk for cardiac complications, including cardiac failure, cardiac wall weakening, and cardiac hypertrophy. What hypothesis could be drawn? What could be discussed?

- The authors observe a statistically significant association between "lung disease" and the development of cardiac problems. Despite the Chagas' disease diagnosis, lung disease can have an independent relationship to the occurrence of right-sided cardiac complications. What hypothesis could be drawn? What could be discussed?

- The authors observe a statistically significant association between "neurological disease" and the development of cardiac problems. Why wasn't anything mentioned? What hypothesis could be drawn? What could be discussed?

- The authors observe a statistically significant association between "thyroid diseases" and the development of cardiac problems. Why wasn't anything mentioned? What hypothesis could be drawn? What could be discussed?

- The authors observe a statistically significant association between the "Charlson comorbidity index" and the development of cardiac problems. What hypothesis could be drawn? What could be discussed?

- The authors observe a statistically significant association between the "female gender" and the development of digestive problems. What hypothesis could be drawn? What could be discussed?

- The authors observe a statistically significant association between "depression" and the development of digestive problems. Female gender is usually associated with a higher incidence of depression. Would the accurate correlation remain with gender, or does depression play a role as a predictor? What hypothesis could be drawn?

- The authors did not observe a statistically significant association between "diverticular disease" and the development of digestive problems. Diverticular disease habitually happens due to the weakening of the intestinal wall. Why couldn't this have played a role in complications?

Reviewer #3: (No Response)

**Editorial and Data Presentation Modifications?**

Reviewer #1: There are a few typographic errors (Benzinadole; neglection; fibilation). Some conceptual issues:

- though BENEFIT trial include long term patient follow-up, it did not include patients with chronic indeterminate Chagas disease, but patients with early cardiac disease manifestations);

- difficult to state that almost all patients with Chagas live in South America (as stated in the introduction), when one has circa 40,000 patients in Spain and around 300,000 in the US.

- parenteral transmission is not a recent form of spread of the disease (as stated in the Introduction), but rather it has been recognised for decades and control measures significantly reduced its impact)

Reviewer #2: - There is a disruption in the manuscript's flow. The "Title" and "Introduction" suggest that the manuscript's primary focus is to determine disease predictors for the development of cardiac and digestive disorders among patients with chronic indeterminate Chagas' disease. However, the center stage of the "Discussion" section is given to an elaborate review of the therapeutic potential of Benznidazole. The "Discussion" should be reworked, and the flow of the manuscript rearranged to reflect what the manuscript describes as its purpose. (Please see more details on the peer review feedback for "Results," "Conclusions" and ""Summary and General Comments").

- Although some grammatical issues can be perceptible to a native speaker of American English, they are mild and do not interfere with the manuscript's comprehension.

- On page 6, line 2, the drug Benznizadole has a typo "Benzinadole"

Reviewer #3: (No Response)

**Summary and General Comments**

Reviewer #1: There is an important need for cohorts with long term follow-up of clinical outcomes in Chagas and the assessment of impact of different predictors of treatment response (including treatment). The study has some limitations which could certainly be addressed with significant revisions and clarifications. It is really unfortunate, however, that the supplementary information provided includes the name of patients and other identifying information. This represent a serious data privacy and ethical issue in implementation of clinical research, which I believe it would be challenging to resolve. This would require editorial consideration.

Reviewer #2: Summary

- The authors' interest in the topic of Chagas Disease and further discern its predictors is a commendable intention. As the authors also go to explain, despite the prevalence of Chagas Disease, especially its indeterminate chronic form, what is often offered to patients is not enough to improve prognosis and disease outcome. Therefore, the study's primary aim to better determine disease predictors is an objective of high relevance, both from research and clinical perspectives. Determining predictors for disease severity provides the opportunity to troubleshoot patient screening by focusing on particular characteristics.

- However, there is a disconnect between what is described as the primary goal of the study (as described in its title, abstract, and introduction) and what the article focus on in its results, and more importantly, its discussion.

- The introduction sets the stage for the importance of the article's primary aim: determining predictors for the development of cardiac and digestive disorders among patients with chronic indeterminate Chagas disease. Per this primary aim, the results describe the statistical significance of a set of characteristics that may play an important role as predictors of disease. However, the discussion falls short in delving into more details on the findings described in the results sections. Instead, it goes to focus on a hypothesized efficacy of the drug Benznidazole as a protective therapy for the cardiovascular outcomes of Chagas' disease.

- Although there is a statistically significant association between the use of Benznidazole and the development of cardiac disorders among patients with chronic indeterminate Chagas Disease, this is a secondary aim of the study. More focus should be given to describing the results section's findings and establishing plausible hypotheses that justify all the statistically significant variables as possible predictors of disease severity. After all, this was said to be the goal of the study.

- Furthermore, despite the Benznidazole's statistical significance, the findings are not sufficient to claim it as an efficacious drug. Additional studies are necessary to support this claim, including a study design that focuses on the outcomes of said drug and considers possible bias, confounders, and effect modifiers that were not fully explored in this study. As it stands, the use of Benznidazole has an association with the outcome, but causality cannot be suggested.

- The study is relevant research that offers new data collected from patients, which can elicit essential discussions in the scientific community. However, when trying to focus on the drug, the authors miss the opportunity to engage in a fruitful discussion on the bulk of the obtained results.

- The statistically significant variables that are possible predictors, and the not statistically significant variables that are discarded as predictors offer a relevant story that should be told, and hypotheses that should be discussed. I encourage the authors to rework the manuscript to establish a more explicit correlation between their aim of identifying predictors of disease outcomes, as they initially described in the "Introduction" and the details focused on the "Discussion" section.

Additional recommendations to the authors:

- The attachment named "renamed_0bb9e.xlsx" contains patients' names and patient record numbers. I strongly suggest that the authors replace the document with a version that preserves the study participants' anonymity. Efforts should be taken to substitute patient identifiers with particular encrypted codes that only the research team could use to identify them.

- On the "Author Summary" section, page 3, lines 3-5; and "Introduction," page 4, lines 6-8, the idea of parenteral transmission through blood transfusion is described, also highlighting the risks of Chagas' disease transmission from the migration from South America to other parts of the world and the subsequent involvement of these individuals in blood transfusion. Although scientifically sound, the concept of the potential risk of South Americans' migration to other parts of the world should be treated with more responsibility in the manuscript. These particular sections of the manuscript could easily be taken out of context when presented to the public at large, reinforcing stereotypes and fueling xenophobia and racism again South American immigrants. The authors should make an effort in the passage's language to enforce the scientific interpretation and avoid any claims that could be taken out of context to propagate discrimination. I understand this was not the authors' intention, but I believe it is an important point to bring to the attention since the readers can be from areas facing different political climates.

Reviewer #3: (No Response)

PLOS authors have the option to publish the peer review history of their article (what does this mean?). If published, this will include your full peer review and any attached files.

Reviewer #1: No

Reviewer #2: No

Reviewer #3: No
---

## [Decision Letter · Decision Letter 1]

27 Jan 2021

Dear Dr. Fortaleza,

Thank you very much for submitting your manuscript "Predictors of development of cardiac and digestive disorders among patients with chronic indeterminate Chagas Disease: emphasis on the impact of Benznidazole therapy." for consideration at PLOS Neglected Tropical Diseases. As with all papers reviewed by the journal, your manuscript was reviewed by members of the editorial board and by several independent reviewers. In light of the reviews (below this email), we would like to invite the resubmission of a significantly-revised version that takes into account the reviewers' comments. 

We cannot make any decision about publication until we have seen the revised manuscript and your response to the reviewers' comments. Your revised manuscript is also likely to be sent to reviewers for further evaluation.

Sincerely,

Christine A Petersen

Deputy Editor

Christine Petersen

Deputy Editor

Reviewer's Responses to Questions

**Key Review Criteria Required for Acceptance?**

**Methods**

-Are the objectives of the study clearly articulated with a clear testable hypothesis stated?

-Is the study design appropriate to address the stated objectives?

-Is the population clearly described and appropriate for the hypothesis being tested?

-Is the sample size sufficient to ensure adequate power to address the hypothesis being tested?

-Were correct statistical analysis used to support conclusions?

-Are there concerns about ethical or regulatory requirements being met?

Reviewer #2: -The objectives of the study were clearly articulated with a clear testable hypothesis. The clarity of the aims has improved since the original submission. This time, two aims were characterized. The first is to assess possible variables/predictors that contribute to the evolution of the indeterminate chronic form of Chagas disease into the disease's cardiac or digestive conditions. The second, to assess the impact that Benznidazole has as a therapeutic intervention in ICF of Chagas disease, focusing on its ability to prevent the progression to cardiac and/or digestive forms of Chagas disease.

- The study design appropriately addresses the stated objectives. A non-concurrent cohort study analyzing data collected from Chagas disease patients being followed prospectively in the outpatient clinic over two decades. The study's data was then retrieved retrospectively from their medical charts.

- The population was clearly described and seemed appropriate for the hypothesis being tested. The study group recruited Chagas' disease patients from the Tropical Diseases outpatient service in Botucatu Medical School, São Paulo State University (UNESP), who presented with the indeterminate chronic form of Chagas disease from the year 1998 through 2018, with an extended follow-up period to December 2019.

The inclusion criteria demanded a Chagas disease diagnosis based on positivity in two serological tests (ELISA and indirect immunofluorescence). Only patients who were positive in both were included in the study. As exclusion criteria, those who presented abnormalities that could be attributed to either cardiac or digestive forms of CD upon admission were excluded from the study, which was an appropriate measure since the group wanted to assess the risk of evolving in these conditions and not recruiting patients who already had them. Patients lost to follow-up were also excluded from the study.

A noteworthy mention is that this data comes from a specific city of Brazil with a particular set of demographic characteristics. Therefore, caution should be taken when discussing the data's generalizability to other populations who don't share all features. However, the study's publication will be interesting to other groups who could then attempt to reproduce the design in different regions of Brazil or country, further assessing similarities and differences in the patterns gathered from other populations.

- No mention is made in the manuscript to describe if the sample size is sufficient and ensure adequate power to address the tested hypothesis. However, the study uses a convenient sample of all the patients with a Chagas Disease diagnosis admitted while presenting the disease's indeterminate chronic form. The study does yield statistical significance.

- Statistical analyses utilized univariate and multivariable Cox regression models, which could be considered appropriate to assess associations between particular variables and outcomes of interest as set by the study's two primary goals.

- There are no current concerns about ethical or regulatory requirements. The study was exempt by the local IRB, and approval information was provided. Also, the breach of patient confidentiality exhibited in the original submission, which contained patients' names and chart numbers, were addressed in the authors' responses to reviewer comments and fixed in the new submission.

Reviewer #3: The authors have addressed many of the comments raised by the reviewers of the original manuscript. The objectives are clearly stated and address critical questions about Chagas disease, but there are flaws in the study design, particularly pertaining to heart disease. The analysis for gastrointestinal disease is more robust. 

1. The authors do not define "abnormalities that could be attributed to chronic Chagas disease" when defining indeterminate Chagas disease; given the high percentage of persons with important cardiovascular comorbidities, there is a real possibility of misclassification. The authors should clearly elucidate which abnormalities were permissible for inclusion in their cohort. The classic definition of indeterminate Chagas disease is normal history, physical exam, chest xray and ECG; the authors could consider a new definition that includes persons with comorbidities.

2. The criteria for selecting which persons received benznidazole are not spelled out. Analysis of persons with and without certain comorbidities suggest bias that favors the effectiveness of benznidazole therapy. For example, Only 9% persons with pre-existing cardiac disease received benznidazole as opposed to almost 20% of those classified as not having cardiac disease upon entry. Similarly, only 15% of those with hypertension received benznidazole as opposed to 22% who did not have hypertension. 

3. The authors do not define the timing of benznidazole therapy. Periods of follow-up after administration of benznidazole may be shorter than the periods of follow-up of persons who did not receive benznidazole. 

4. There is likely to be misclassification of outcomes. For example, isolated BASRE (in the spread sheet) is defined as a specific endpoint for Chagas disease. (This is much less specific than right bundle branch block, but has high specificity when seen with RBBB.) It is well known that isolated BASRE (also known as left anterior hemiblock or anterior fascicular block) is common among person with hypertension and its prevalence increases with age. In the Excel sheet there are 21 persons who developed BASRE; 13 had preexisting hypertension (62%); only 8 persons of 163 without preexisting hypertension developed BASRE (5%); all 3 persons who had complete left bundle branch had preexisting hypertension.(note that BASRE + LBBB =the 24 persons with LBBB in table 3).Moreover, the persons who developed BASRE are older at endpoint (average 67.9 years) than those who did not develop BASRE (61.9 years).

Note--would not lump LBBB and LAH into one category.

**Results**

-Does the analysis presented match the analysis plan?

-Are the results clearly and completely presented?

-Are the figures (Tables, Images) of sufficient quality for clarity?

Reviewer #2: - The analysis presented matches the analysis plan.

- Tables 1 and 2: Classically, subjects characteristics demographic information is shown in table 1. In other words, invert the order of the current tables 1 and 2.

- Table 3: please see suggestions in the "Editorial and Data Presentation Modifications" section. Table 4: no issues.

- Tables 5 and 6: In the multivariable analysis column, are the lines in blank not tested or just not significant. If tested, please include them in the table.

- Emphasis was given to portraying the results through tables and figures. However, having some highlights in the results could help the reader be more familiar with the results that the authors considered to be most relevant.

Reviewer #3: Presentation is clear

**Conclusions**

-Are the conclusions supported by the data presented?

-Are the limitations of analysis clearly described?

-Do the authors discuss how these data can be helpful to advance our understanding of the topic under study?

-Is public health relevance addressed?

Reviewer #2: - The discussion section has a statement "Data from literature agree with our findings," which findings are the ones referred to by the authors?

- The first part of the discussion focuses on mentioning other studies but fails to appropriately discuss the results found in table 4, table 5, table 6, figure 1, and figure 2.

- The discussion only briefly introduces the discussion on the cardiovascular form of Chagas disease but only focuses on the systemic arterial hypertension component of the results. The discussion should expand on the interpretation of the cardiovascular findings from the results section.

- The discussion doesn't delve into the digestive form of the disease, which was one of the study objectives. It only mentioned the digestive complication concerning the use of Benznidazole.

-The authors discuss how these data can be helpful to advance our understanding of Chagas disease.

- The public health relevance of the manuscript is discussed in the abstract and conclusion.

- The discussion and conclusion seem to focus on Benznidazole and miss the opportunity to discuss the results on the predictors for the cardiovascular and digestive forms of Chagas Disease.

Reviewer #3: Several conclusions, especially regarding benznidazole therapy, are not supported by the data presented for reasons above. The authors nicely address other aspects of the study in their conclusions.

**Editorial and Data Presentation Modifications?**

Reviewer #2: 1- A minor suggestion is the change of table order. Classically, subjects’ characteristics and demographic information are shown as the “table 1”. In other words, I suggest inverting the order of the current tables 1 and 2.

2- I understand the attempt of blacking out the cells of table 1 to make it more visually appealing, but I believe it is not entirely intuitive for all readers. I would suggest reworking the visual aspect of the current table 1. Some suggestions would be having two columns, the second column explaining when the assessment is done. Another option is to substitute the black and whites with “Yes” and “No.”

3- I appreciate that the title now has a mention of Benznidazole therapy. Since the Benznidazole therapy is a second aim and not a focus of the first aim, I would suggest that rather than saying “emphasis on,” the authors could adopt “and.” The authors also call the study disease ICF “indetermined chronic form,” but in the title, they display a different order of the words like “chronic indeterminate Chagas.” I would then suggest adapting the disease name in the title to match the rest of the manuscript. The proposed title would then read: “Predictors of development of cardiac and digestive disorders among the indeterminate chronic form of Chagas Disease and the impact of Benznidazole therapy.”

4- A few times in the article, the disease of interest is described as “indetermined chronic form (ICF)”; other times, it is called “chronic indeterminate,” including in the tables. I suggest using only one form to keep consistency.

5- Most of the work that needs addressing is in the discussion section. The study's primary aims are to identify predictors that would influence the progression of Chagas disease in its indeterminate chronic form to either the cardiac form of the disease or digestive condition and the impact of Benznidazole therapy as a possible protector. The discussion spends its first few paragraphs discussing other publications. Then, it only highlights one of the results pertaining to the cardiac form of the disease and only discusses the digestive form related to the use of Benznidazole.

The discussion should be used as an opportunity to highlight the results and how they address the study’s objectives, including the association with possible predictors for the cardiac and digestive forms of the disease, and then if Benznidazole had an impact and if that was protective.

6- Other minor changes:

5.1- Doença de Chagas can be represented as “Chagas Disease,” without the need for an apostrophe to designate possession of a noun ended with “s.” The manuscript has a recurrent variation as “Chagas’ Disease,” which could be modified to “Chagas disease.”

5.2- Table 2 note says, “Statically,” please modify to “Statistically.”

4.3- Table 3 designates ventricular disfunction as “DV.” I suggest adapting to “VD” to match the other acronyms in the table.

5.4- The table has a typo “Dilateed,” please fix it to “Dilated.”

5.5- Please review the manuscript for other small typos or words kept in Portuguese.

Reviewer #3: Would suggest reanalysis of this rich data set to look at the importance of comorbidities on the evolution of chronic Chagas heart disease as well as characteristics of Chagas disease in older populations. The raw data themselves are important! A purely descriptive paper about evolution of Chagas disease in older persons and persons with comorbidities would be a terrific contribution to the literature. Would deemphasize data on benznidazole, which are misleading. 

The analysis of gastrointestinal disease is sounder.

**Summary and General Comments**

Reviewer #2: The study sets to determine predictors for the development of cardiac and digestive disorders among patients with chronic and indeterminate Chagas Disease. The study's purpose is outstanding. Such determinants can enlighten clinicians on the particular variables that are most relevant for monitoring patients and ascertain the medical team's conduct. The study also wants to assess Benznidazole therapy outcomes and if such drug is associated with a protective effect to prevent the development of the cardiac or digestive form of the disease. The findings would also have the potential to establish a new hypothesis to incentivize new research avenues to study Chagas Disease.

In this revised manuscript, the authors addressed most of the reviewers' concerns with the original submission. This research is worthy of publication, but the authors are having some issues in conveying the importance of their findings. Interesting data is presented in the results section, but not much of it is being portrayed or highlighted in the discussion section.

Most of the manuscript only requires some fine adjustments. However, the discussion and conclusion are what most need to work.

Most of the information is relayed with tables and figures rather than the text in the results section. This is an understandable strategy, which would also avoid the redundancy of having the same thing stated in text and tables. However, this leaves the discussion as a stage where the authors should highlight their findings and what they mean. However, the initial part of the discussion focuses on reiterating information from other publications. The middle part of the discussion mentions systemic arterial hypertension as a highlight of the cardiac form of Chagas disease but makes no further mention of other predictors. And the only comment on the digestive form of the disease is to outcomes related to the use of Benznidazole.

The authors are on the right track, and their work is worthy of publication. However, they need to tie the discussion and conclusion back to the goals they described in the introduction and methods. With that said, please make sure the discussion section is highlighting the findings pertaining to predictors of the cardiac form of Chagas disease, highlighting predictors of the digestive form, and only then discusses the findings as they relate to Benznidazole. If the authors address the minor concerns and make sure the discussion correlates their findings to their aims, I believe the manuscript will likely be ready for publication.

Reviewer #3: The strength of the paper is the study population: persons with indeterminate Chagas disease, many of who have important comorbidities and are older than subjects of earlier studies of the natural history of Chagas disease. The study has the potential to fill important gaps in our understanding of chronic Chagas disease. Would suggest reanalysis of data looking at outcomes as a function of comorbidities and analyze data without (as well as with) benznidazole treatment.

PLOS authors have the option to publish the peer review history of their article (what does this mean?). If published, this will include your full peer review and any attached files.

Reviewer #2: No

Reviewer #3: No
---

## [Decision Letter · Decision Letter 2]

6 Jun 2021

Dear Dr. Fortaleza,

Thank you very much for submitting your manuscript "Predictors of development of cardiac and digestive disorders among patients with indeterminate chronic Chagas Disease, with remarks on the potential impact of Benznidazole therapy" for consideration at PLOS Neglected Tropical Diseases. As with all papers reviewed by the journal, your manuscript was reviewed by members of the editorial board and by several independent reviewers. The reviewers appreciated the attention to an important topic. Based on the reviews, we are likely to accept this manuscript for publication, providing that you modify the manuscript according to the review recommendations. 

Although the manuscript is improved, there are still some problematic concerns about the potential of misclassification bias regarding cardiac disease in the design of the assessment of benznidazole treatment. Please see the comments from reviewer 3.

The analysis of impact of benznidazole on progression of heart disease is severely flawed, and confounding factors have not been sufficiently addressed. If possible, better adjustment for these factors would help, otherwise they must be discussed as flaws in the study design and the findings of the cardiac patient analyses reconsidered in that light.

Sincerely,

Christine A Petersen

Deputy Editor

Christine Petersen

Deputy Editor

Although the manuscript is improved, there are still some problematic concerns about the potential of misclassification bias regarding cardiac disease in the design of the assessment of benznidazole treatment. Please see the comments from reviewer 3.

The analysis of impact of benznidazole on progression of heart disease is severely flawed, and confounding factors have not been sufficiently addressed. If possible, better adjustment for these factors would help, otherwise they must be discussed as flaws in the study design and the findings of the cardiac patient analyses reconsidered in that light.

Reviewer's Responses to Questions

**Key Review Criteria Required for Acceptance?**

**Methods**

-Are the objectives of the study clearly articulated with a clear testable hypothesis stated?

-Is the study design appropriate to address the stated objectives?

-Is the population clearly described and appropriate for the hypothesis being tested?

-Is the sample size sufficient to ensure adequate power to address the hypothesis being tested?

-Were correct statistical analysis used to support conclusions?

-Are there concerns about ethical or regulatory requirements being met?

Reviewer #2: - The objectives of the study were articulated with a clear testable hypothesis. The clarity of the objectives has improved in this submission.

- The study design is appropriate to addresses the stated objectives. A causality determination is not possible with the model, so it is essential to designate the results as “associations” between exposure and outcome. The authors have appropriately addressed such concerns in this submission.

-The population of interest was clearly described. It was characterized as those who had the diagnosis of indeterminate chronic form of Chagas Disease in a particular Brazilian city. Both Elisa and Indirect immunofluorescence confirmed the diagnosis of indeterminate chronic form of Chagas Disease. The hypothesis was clearly described to review these patients’ charts for the progression of cardiac or digestive forms of Chagas disease and statistical analysis of possible predictors. In addition, for those who used Benznidazole therapy, the use of the medication was also assessed as a possible predictor that may have deterred the progression of the disease to cardiac or digestive forms.

- The study uses a convenient sample of all the patients with a Chagas Disease diagnosis who were admitted while presenting the disease’s indeterminate chronic form (ICF), and the study does yield statistical significance. The authors also addressed the concern from the previous submission about power, both in response to reviewers and new manuscripts, and described appropriate calculations.

- Statistical analyses utilized univariate and multivariable Cox regression models, which could be considered appropriate to assess associations between particular variables and the outcomes of interest.

- The authors have adjusted the recommendation of caution when describing the data’s generalizability to other populations who do not share all features of their study population.

- The study does not present concerns about ethical or regulatory requirements being met. It followed

principles from the Helsinki declaration and was approved by their local Committee for Ethics in Human

research.

The following points deserve clarification in the Methods section:

1- Define Benznidazole criteria if available: The authors mention that “Benznidazole was used when indicated by the attendant doctor, based on the Brazilian Ministry of Health recommendations,” however, other than pointing the reader towards the reference, the clinical criteria for these patients’ eligibility for Benznidazole was not defined. Please outline the characteristics used by the team to define eligibility. Since the Benznidazole therapy was studied as a variable of preventive association with the onset of cardiac and digestive forms of Chagas, it is important to understand why certain patients may have been eligible for the drug while others were not under the Brazilian guidelines. Not all patients are eligible for antiparasitic treatment, and the reasoning for prescription or contraindication may also be due to characteristics that may interfere with the statistical analysis or serve as confounders. However, considering the retrospective nature of the study and that data were from charts, it is possible that the authors did not have access to the physicians’ reasoning and criteria, just if Benznidazole had been used or not. If authors had no access to the criteria utilized by the attending physicians, this should be specified in the methods. 

2- Better characterize which parameters were used for exclusion criteria: “Those who presented abnormalities that could be attributed to either cardiac or digestive forms of CD upon admission were excluded from our study.”, “our inclusion criterium was the admission with ICF, defined as “serological positivity for CD, without presenting any cardiac or digestive abnormalities that could be attributed to that disease.” Please characterize which parameters were considered for the exclusion of participants. Was it based solely on ECG cardiological findings like outcomes in table 3, or did it consider chest radiography, echocardiogram like table 1? For example, for the cardiologic form of CD: “dilated cardiomyopathy, congestive heart failure, arrhythmias, cardioembolism, thromboembolic events, stroke….”

3- Define “heart disease”: A point brought up in previous submissions and not yet fully addressed is what is being considered “heart disease.” In some tables, this is not specified. In others, it is suggested to be categorized as coronary diseases “Heart diseases not attributed to Chagas Disease were restricted to coronary syndrome, with or without myocardial infarction.” Please clarify the author’s definition of “heart diseases” in the method section.

Reviewer #3: Objectives are clearly stated and overall design appropriate.

Methods--this is a retrospective study; informed consent was not obtained on entry to the study and data were reviewed retrospectively.

Study design is appropriate for observational study. Population clearly described.

Problem with study is emphasis on evaluation of impact of benznidazole therapy, which is compromised by confounding. For example, just looking at the variable "heart disease", benznidazole was given to a lower proportion of persons who had preexisting heart disease on entry to the study (5/56) than those who didn't have heart disease (64/323); 28/56 of persons with preexisting heart disease had progression of heart disease; only 59/323 persons without preexisting heart disease developed heart disease. A quick look at table 2 shows a number of other comorbidities that predispose to heart disease (hypertension, diabetes) that are disproportionately distributed between the 2 outcomes and are also likely sources of confounding.

 Another major problem is misclassification of outcome (table 3): in this study LBBB is a more frequent outcome than RBBB; while LBBB is common in persons with other forms of heart disease (hypertension, LVH, aortic stenosis, etc) it is seen in a minority of persons with Chagas disease as the only cause of heart disease.

Statistical analysis appropriate; no ethical concerns.

**Results**

-Does the analysis presented match the analysis plan?

-Are the results clearly and completely presented?

-Are the figures (Tables, Images) of sufficient quality for clarity?

Reviewer #2: - The analysis presented matches the analysis plan.

- The authors have expanded the analysis plan as suggested in the previous submission.

- The authors have incorporated the suggested changes from the previous submission, including more appropriate use and description of their tables and images.

Reviewer #3: 1.The analytic methods are appropriate.

2.Results are clearly presented but not completely presented. Results of simple analysis showing treatment status of persons with pre-existing conditions and those without would allow the reader assess possible confounding.

3.The tables and figures are clear.

**Conclusions**

-Are the conclusions supported by the data presented?

-Are the limitations of analysis clearly described?

-Do the authors discuss how these data can be helpful to advance our understanding of the topic under study?

-Is public health relevance addressed?

Reviewer #2: -The conclusions have been made more evident in this submission. In addition, the authors have expanded the discussion to other predictors rather than the extended focus on Benznidazole therapy that had been given in previous submissions.

-Most limitations were acknowledged, and additional quantifications were introduced to address some of these imitations.

- The authors discuss how these data can be helpful to advance our understanding of indeterminate chronic Forman acknowledge that proper clinical trials are necessary to address the suggested benefit from Benznidazole.

- The public health relevance of the manuscript's finding is addressed.

Reviewer #3: 1.Conclusions about impact of comorbidities on progression of heart disease are sound; conclusions about impact of benznidazole therapy are not. 

Conclusions about impact of benznidazole on progression of gastrointestinal disease are on more solid grounds because of less potential confounding and more precise definitions of outcomes.

Rather than defining their population as persons in the indeterminate stage of Chagas disease, the authors should distinguish between persons without evidence of heart disease and persons with evidence of heart disease other than Chagas disease.

2. The authors do not discuss potential confounding and misclassification.

3. Discussion of importance of comorbidities on progression of Chagas disease is appropriate.

4. Conclusion that controlled trials are needed is sound.

**Editorial and Data Presentation Modifications?**

Reviewer #2: 1- Please refer to suggestions made on methods section feedback.

2- The authors define the criteria for outcomes in the "Results" section. These are often described in "Methods," the switch could contribute to clarity.

3- The manuscript contains minor typographic errors throughout the text. For example, some words are in Portuguese, such as "áreas" in the tables, ICF is once referred to as IFC, missing spaces, missing periods, etc. Please review the entire manuscript for typos.

Reviewer #3: Manuscript is well-written could benefit from minor editing for clarity

**Summary and General Comments**

Reviewer #2: The authors have provided significant improvements that have brought more clarity to their manuscript.

Reviewer #3: Strengths of the study are the unique patient population of persons with T cruzi infection--they are older and have a high rate of well defined cardiovascular comorbidities and risk factors than persons in many other studies of Chagas disease. I believe that this is the great strength of the data set and the paper, and the authors' discussion of this is a strength.

Weakness of the study is the emphasis on impact of benznidazole therapy on outcome of cardiovascular disease. Would recommend not making this an objective of the study; ok to present the data and observations but would make the point that conclusions cannot be drawn because of potential confounding; would also make clear that case definition (of indeterminate form that includes cardiac conditions not felt to be from Chagas disease) is not conventional, and that certain cardiac outcomes (eg, LBBB) are more typical of other heart diseases than Chagas heart disease.

The analysis of impact of benznidazole treatment on gastrointestinal outcome is more robust because there is less potential confounding and the outcomes are more precise.

PLOS authors have the option to publish the peer review history of their article (what does this mean?). If published, this will include your full peer review and any attached files.

Reviewer #2: No

Reviewer #3: No

Figure Files:

Data Requirements:

Reproducibility:

References

---

## [Decision Letter · Decision Letter 3]

26 Jul 2021

Dear Dr. Fortaleza,

We are pleased to inform you that your manuscript 'Predictors of development of cardiac and digestive disorders among patients with indeterminate chronic Chagas Disease.' has been provisionally accepted for publication in PLOS Neglected Tropical Diseases.

Best regards,

Christine A Petersen

Deputy Editor

Christine Petersen

Deputy Editor

Please remove accents from tables to appear in this English print journal.

Reviewer's Responses to Questions

**Key Review Criteria Required for Acceptance?**

**Methods**

-Are the objectives of the study clearly articulated with a clear testable hypothesis stated?

-Is the study design appropriate to address the stated objectives?

-Is the population clearly described and appropriate for the hypothesis being tested?

-Is the sample size sufficient to ensure adequate power to address the hypothesis being tested?

-Were correct statistical analysis used to support conclusions?

-Are there concerns about ethical or regulatory requirements being met?

Reviewer #2: - The objectives of the study were properly articulated with a clear testable hypothesis stated. Following reviewers suggestions, the emphasis has switched from the outcomes of benznidazole therapy to the prediction of cardiac and digestive forms of Chagas Disease.

- The study design appropriately addresses the objectives.

- The population is properly described, constituting a retrospective review of charts on with 20 years worth of information from patients in a rural area of Brazil.

- The sample size is appropriate to address the study goals and followed by proper statistical analysis.

- There are no concerns about ethical or regulatory requirements and supplemental patient data has been anonymized.

- The selection criteria has been expanded to follow the reviewers recommendations.

Reviewer #3: Methods ok though analysis of benznidazole effect would be more robust if authors looked separately at persons who had heart disease, persons who had hypertension and persons who had neither. The authors said that they did this in table 7, but if I'm reading table 2 and 7 correctly they only analyzed persons with heart disease.

Definition of cardiac outcome (LBBB) remains flawed

**Results**

-Does the analysis presented match the analysis plan?

-Are the results clearly and completely presented?

-Are the figures (Tables, Images) of sufficient quality for clarity?

Reviewer #2: - The results meet the analysis plan, and results have been expanded to feature sensitive analysis and incorporated tables, figures and texts addressing the reviewers concerns.

- Additional figures are offered by authors (figure 1, figure 2, and figure 3), and these may be clearer to viewers if additional description is provided below the figures.

Reviewer #3: Presentation, figures etc clear

Table 2 : 57% of persons had hypertension, yet the analysis (results in table 7) seems to include persons with hypertension.

**Conclusions**

-Are the conclusions supported by the data presented?

-Are the limitations of analysis clearly described?

-Do the authors discuss how these data can be helpful to advance our understanding of the topic under study?

-Is public health relevance addressed?

Reviewer #2: - Following the reviewers recommendations, the authors seem to have given appropriate focus to the factors that may influence the development of cardiac and digestive forms of Chagas disease rather than a disproportional focus on the incidental finding of possible Benznidazole therapy benefits.

- Benznidazole is presented as a secondary finding worth of further exploration rather than the primary focus of the manuscript, which has been a concern since its first submission.

- The authors acknowledge the limitations of their findings and address the steps they have taken to address them. Particularly, the authors recognize the limitation in claiming that the findings would have been sufficient to point Benznidazole as a prevention to cardiac form of Chagas Disease. Instead, they mostly attain to the statistical analysis and point out the limitations of such claim, including that preexisting cardiac conditions or independent risk factor for cardiovascular diseases could have also served as confounders. By acknowledging the limitations, and better describing that statistical approach to address them, the authors seem to address most of our concerns as to the way the Benznidazole data was originally presented.

- The authors also go on to explain research avenues that should be pursued in the future to further explore their current findings, bringing light to the public health relevance of this manuscript.

Reviewer #3: Overall paper is improved, but conclusions re benznidazole can be misleading.

I won't go through different variables again, but let's look only at hypertension. 57% of participants had hypertension on entry. 73.5% of persons who had progressive heart disease had hypertension at baseline. 24/87 persons who progressed to "Chagas heart"disease progressed to having LBBB, an unusual marker of Chagas disease and a common marker of hypertensive heart disease. How many people with hypertension received benznidazole, how many did not?

Given large numbers, why not look at subgroups: eg persons with hypertension, persons with heart disease, persons with neither. This could be a contribution to the benznidazole question--eg, do persons who have pre-existing heart disease or hypertension benefit from benznidazole. It would also contribute to the still unknown impact of benznidazole for persons who have no cardiovascular disease. This was attempted in table 7, but unless I'm misreading the paper, persons with hypertension are not included

**Editorial and Data Presentation Modifications?**

Reviewer #2: - If figures 1-3 contain an explanation under them, they would be clearer for independent viewing of the reader.

- Manuscript contains some minor typos, for example: a random "~ç" on Methods, "Living in rural áreas" on tables 2 and 5, "SolidMalignancy" on table 2, "se above" on Discussion. These can be easily addressed when formatting the manuscript for publication

Reviewer #3: Would downplay suggestion that benznidazole is beneficial.

**Summary and General Comments**

Reviewer #2: The authors addressed most of the concerns the reviewers have posed to their previous submissions, especially by toning down the interpretation of the data that seemed to focus on their incidental finding of Benznidazole therapy and protective potential. The authors expanded their acknowledgement of limitation, steps taking to reduce them and future research avenues that could further clarify the findings.

Reviewer #3: Potentially an important contribution to literature about natural history of Chagas disesase with and without important cardiovascular comorbidities. Weakness are conclusions re effect of benznidazole.

PLOS authors have the option to publish the peer review history of their article (what does this mean?). If published, this will include your full peer review and any attached files.

Reviewer #2: No

Reviewer #3: No

---

## [Editor Report · Acceptance letter]

5 Aug 2021

Dear Dr. Fortaleza,

We are delighted to inform you that your manuscript, "Predictors of development of cardiac and digestive disorders among patients with indeterminate chronic Chagas Disease," has been formally accepted for publication in PLOS Neglected Tropical Diseases.

Best regards,

Shaden Kamhawi

co-Editor-in-Chief

Paul Brindley

co-Editor-in-Chief
